# Cloud Cover throughout All the Paddy Rice Fields in Guangdong, China: Impacts on Sentinel 2 MSI and Landsat 8 OLI Optical Observations

**Rui Jiang** [1,2,3,4,5], **Arturo Sanchez-Azofeifa** [2], **Kati Laakso** [2], **Yan Xu** [1], **Zhiyan Zhou** [1,3,4,5,*,†], **Xiwen Luo** [1,3,4,5], **Junhao Huang** [1,3,4,5], **Xin Chen** [6] **and Yu Zang** [1,3,4,5,†]

1   College of Engineering, South China Agricultural University/Guangdong Engineering Research Center for Agricultural Aviation Application (ERCAAA), Guangzhou 510642, China; ruilo.jiang@stu.scau.edu.cn (R.J.); autoxuyan@stu.scau.edu.cn (Y.X.); xwluo@scau.edu.cn (X.L.); 201621180609@stu.scau.edu.cn (J.H.); zangyu@scau.edu.cn (Y.Z.)
2   Centre for Earth Observation Sciences (CEOS), Department of Earth and Atmospheric Sciences, University of Alberta, Edmonton, AB T6G 2E3, Canada; gasanche@ualberta.ca (A.S.-A.); laakso@ualberta.ca (K.L.)
3   National Center for International Collaboration Research on Precision Agricultural Aviation Pesticides Spraying Technology (NPAAC), Guangzhou 510642, China
4   Key Laboratory of Key Technology on Agricultural Machine and Equipment, South China Agricultural University, Ministry of Education, Guangzhou 510642, China
5   Guangdong Provincial Key Laboratory of Agricultural Artificial Intelligence (GDKL-AAI), Guangzhou 510642, China
6   Department of Land Resources Management, College of Land Science and Technology, China Agricultural University, Beijing 100193, China; chenxin1992@cau.edu.cn
*   Correspondence: zyzhou@scau.edu.cn; Tel.: +86-135-6002-6139
†   Zhiyan Zhou and Yu Zang contributed equally to this work.

**Abstract:** Cloud cover hinders the effective use of vegetation indices from optical satellite-acquired imagery in cloudy agricultural production areas, such as Guangdong, a subtropical province in southern China which supports two-season rice production. The number of cloud-free observations for the earth-orbiting optical satellite sensors must be determined to verify how much their observations are affected by clouds. This study determines the quantified wide-ranging impact of clouds on optical satellite observations by mapping the annual total observations (ATOs), annual cloud-free observations (ACFOs), monthly cloud-free observations (MCFOs) maps, and acquisition probability (AP) of ACFOs for the Sentinel 2 (2017–2019) and Landsat 8 (2014–2019) for all the paddy rice fields in Guangdong province (APRFG), China. The ATOs of Landsat 8 showed relatively stable observations compared to the Sentinel 2, and the per-field ACFOs of Sentinel 2 and Landsat 8 were unevenly distributed. The MCFOs varied on a monthly basis, but in general, the MCFOs were greater between August and December than between January and July. Additionally, the AP of usable ACFOs with 52.1% (Landsat 8) and 47.7% (Sentinel 2) indicated that these two satellite sensors provided markedly restricted observation capability for rice in the study area. Our findings are particularly important and useful in the tropics and subtropics, and the analysis has described cloud cover frequency and pervasiveness throughout different portions of the rice growing season, providing insight into how rice monitoring activities by using Sentinel 2 and Landsat 8 imagery in Guangdong would be impacted by cloud cover.

**Keywords:** Sentinel 2; Landsat 8; paddy rice fields; cloud cover; cloud-free observations

## 1. Introduction

Rice is a staple crop for more than half of the world's population, especially in the Asian regions [1]. China is the largest rice-producing country while still facing the challenge of rapid food requirement [2]. Therefore, it is crucial to manage rice production in a sustainable manner for alleviating food insecurity [3]. Efficient and timely monitoring

of rice growth is an important aspect of high-quality and sustainable agricultural management [4]. Precision nutrient management (PNM) strategies of rice can help prevent diseases, optimize nutrient use, improve rice yield, and reduce environmental pollution caused by excessive nitrogen use, etc. [5].

Vegetation index inversion of spectral information from optical remote sensing platforms was considered to be a promising and convenient method to contribute to the PNM [6–8]. In recent decades, due to the unique advantages of stable observation time and solar radiation, as well as relatively large coverage, earth-orbiting optical satellite sensors-acquired imagery has been proven and widely used for estimating rice growth by using vegetation indices (Vis), such as normalized difference vegetation index (NDVI) [9], soil-adjusted vegetation index (SAVI) [10], and enhanced vegetation index (EVI) [11], etc. Research has revealed that, in agriculture, the significant changes in crop biomass can occur within a week, which implied that efficient management of crops (e.g., rice) requires high-frequency use of remotely sensed data [12]. However, even if there are many off-the-shelf satellite sensors which have a revisit capability up to 5 days (Sentinel 2A/2B Multispectral Instrument (MSI), combined; Harmonized Landsat 8 Operational Land Imager (OLI) and Sentinel 2 MSI, combined, HLS M30 product), the obscuration by clouds still limits the efficient use of the derived VIs in rice management [13,14]. It is well known that cloud cover has been an uncontrollable natural factor that hinders the effective use of satellite imagery [15,16].

The problem of the extent and intensity of cloud cover is always a challenge and varies distinctively worldwide [17]. Several pioneering studies have assessed the changes in cloud cover at high spatiotemporal resolution by using cloud products estimated from Moderate Resolution Imaging Spectroradiometer (MODIS) data, and it is well recognized that the global cover increased over recent decades; it is estimated that the global average annual cloud cover ranges from 66–70% [18]. In addition, with the influence of natural and human activities, climate change is altering the amount of cloud cover globally and, as such, climate change can also affect the usability of satellite data [19]. For the optical satellite sensors, only the cloud-free observations can be used for the inversion of VIs and, thus, scientific nutrient management of rice [20,21]. In the study of cloud-related observations of optical satellite sensors, Mercury et al. evaluated the effect of global cloud cover on optical satellite observation opportunities with a case of HyspIRI (Hyperspectral and Infra-Red Imager) [22]. It has been reported that, globally, Landsat ETM+ scenes are 35% cloud contaminated [23,24]. Asner reported cloud cover probability with a spatially explicit analysis of cloud cover in the Landsat archive of Brazilian Amazonia from 1984 to 1997, and suggested monthly and annual observations are limited in the basin [25]. Laborde et al. reported the spatial distribution of cloud-free observations (CFO) and cloud-free months (CFM) for the whole period (October 2013–September 2014) on the area of South East Asia (SEA) delimited by the Kingdom of Cambodia, the Lao People's Democratic Republic, and the Kingdom of Thailand, indicating that the spatial distribution of CFO is not spatially homogeneous, and the cloudiness is a major obstacle to optical remote sensing in tropical regions [26]. Xiao et al., reported a comprehensive analysis of cloud cover for Landsat 8 OLI sensor data over China (January 2013–October 2016) and suggested that cloud cover probability analysis is a fundamental prerequisite to land-cover change and earth system process studies in these regions [17]. Cloudiness, therefore, usually varies with geographical location, optical satellite sensors, equatorial zone, year, season, month, day, and even hour, and is greater at high latitudes than at mid-latitudes [27,28].

Although some studies reported the impact of cloud cover on optical satellite observations, it is not difficult to find that previous studies have mainly focused on the analysis at the observation footprint level, and have not evaluated special crops or croplands. To the best of our knowledge, uncertainties in optical remote sensing data due to the frequent influence of clouds and cloud shadows makes it difficult or sometimes impossible to acquire enough cloud-free images during the paddy rice growth stages, especially in cloudy and rainy tropical or subtropical regions. Few studies have attempted to quantify the effects of

cloud cover on satellite-acquired imagery in the context of rice monitoring. On the other hand, few studies have evaluated the effect of cloud cover on Sentinel 2 MSI scenes to date.

In the context of frequent requirements for observing VIs for rice, it is necessary to determine that spatial distribution of the cloud cover on paddy rice fields varies throughout the rice growing season, and in turn impacts optical data acquisitions of the paddy rice fields. To this end, we quantitatively evaluated the cloud restricted availability of Sentinel 2 MSI and Landsat 8 OLI by creating the satellite-based annual total observations (ATOs), annual cloud-free observations (ACFOs), and monthly cloud-free observations (MCFOs) maps for per-field of all the paddy rice fields in Guangdong Province, China. This study contributes to providing useful guidance in the aspect of Landsat 8 OLI and Sentinel 2 MSI data source for monitoring rice in Guangdong.

## 2. Materials and Methods

### 2.1. Study Areas

The study areas were all the paddy rice fields in Guangdong (APRFG, Figure 1), which features a subtropical climate with an annual mean temperature of 22.3 °C and an annual precipitation of 1777 mm [29]. The province has a total cropland area of 44,933 km$^2$, among which paddy rice fields account for 19,873 km$^2$ according to 2008 statistical data of local agriculture department [30]. The abundant precipitation and the average frost-free period of 335–360 days contribute to rice production in the region, making Guangdong one of the major rice-producing provinces in China [31,32]. Guangdong is also an economically developed province, and research on rice breeding and management are also well advanced in China [33]. The application of the optimized management practice in Guangdong has significantly improved the nitrogen (N) use efficiency (NUE), and reduced N fertilizer use by 9.5% compared to conventional farmer's practice [34]. In-time nitrogen excess or deficiency monitoring is an important aspect of rice nutrient management in modern agriculture. Therefore, the VIs obtained by optical satellite sensors played an important role in monitoring the rice growth in this region.

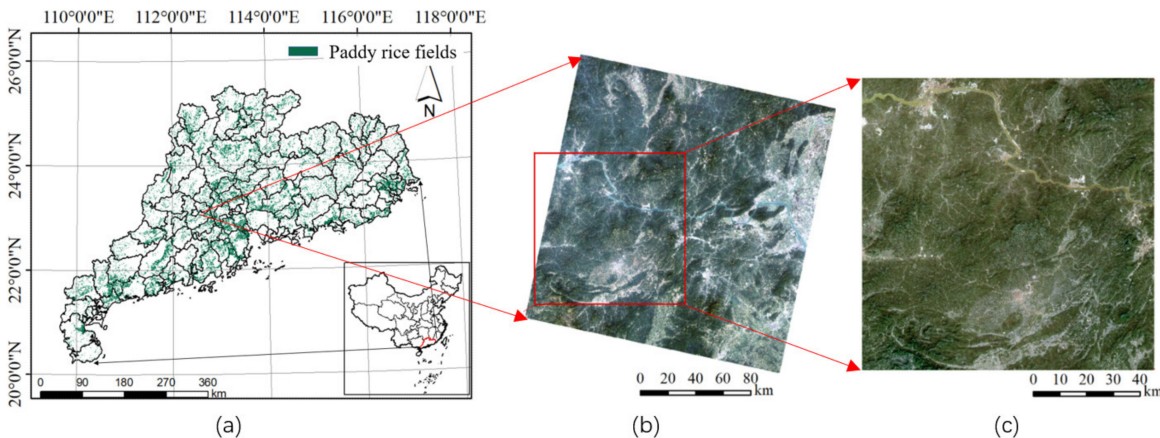

**Figure 1.** Location of study areas: (**a**) the spatial distribution of all the paddy rice fields in Guangdong (APRFG), China; (**b**,**c**) represent Landsat 8 and Sentinel 2 true color image of a subset of the APRFGP taken on 13 May 2019, respectively.

### 2.2. Data Preparation and Processing

The latest APRFG shapefile depicting the spatial distribution of all the paddy rice fields in Guangdong was taken from the Resource and Environment Data Cloud Platform of the Institute of Geographical Sciences and Natural Resources of the Chinese Academy of Sciences (http://www.resdc.cn/, accessed on 1 June 2021), which contains 25,540 1 km × 1 km grid cells that indicate the classified paddy rice fields.

In this study, two data sources, Sentinel 2 MSI imagery (2017–2019) and Landsat 8 OLI imagery (2014–2019) were used for creating the ATOs, ACFOs, and MCFOs maps over

APRFG. The characteristics and the commonly used VIs-related visible and near-infrared wavelengths of Sentinel 2 MSI (https://sentinel.esa.int/, accessed on 1 June 2021) and Landsat 8 OLI (https://landsat.usgs.gov/, accessed on 1 June 2021) were shown in Table 1. Google Earth Engine (GEE), a cloud-based platform, was used to facilitate the management and processing of satellite data [35]. Due to the different launch dates of the Sentinel 2A and Sentinel 2B satellites, the former's images cover October 2015 to December 2019, while the latter's images cover August 2017 to December 2019; therefore, combined Sentinel 2 data (2A and 2B) are only available from 2017 onwards. When compiling the Landsat 8 cloud-related maps, data from 2014–2019 were used, and 2013 was omitted due to possible instability resulting from the 90-day commissioning stage following the satellite launch in 2013 [36]. All the Sentinel 2 (2017–2019) and Landsat 8 (2014–2019) available images that covered the areas of the APRFG were collected by using the GEE's boundaries and date filtering algorithms [37]. The number of available images for Sentinel 2 was 2133 (2017), 4176 (2018), and 4390 (2019), and for Landsat 8 was 284 (2014), 285 (2015), 272 (2016), 292 (2017), 290 (2018), and 255 (2019).

**Table 1.** Partial characteristics of the Sentinel 2 MSI and Landsat 8 OLI.

| Platform | Sensors | Bands | Band Wavelength | Width | Ground Sampling Distance | Revisit Period | Swath |
|---|---|---|---|---|---|---|---|
| Sentinel 2 | Multispectral Instrument (MSI) | B2 | 496.6 nm (S2A) 492.1nm (S2B) | 20 nm | 10 m | 5 days (S2A + S2B at equator); 2–3 days (S2A + S2B mid-latitudes) | 290 km |
| | | B3 | 560 nm (S2A) 559 nm (S2B) | 20 nm | 10 m | | |
| | | B4 | 664.5 nm (S2A) 665 nm (S2B) | 20 nm | 10 m | | |
| | | B8 | 835.1 nm (S2A) 833 nm (S2B) | 20 nm | 10 m | | |
| Landsat 8 | Operational Land Imager (OLI) | B2 | 450–515 nm | 65 nm | 30 m | 16 days | 185 km |
| | | B3 | 525–600 nm | 75 nm | 30 m | | |
| | | B4 | 630–680 nm | 50 nm | 30 m | | |
| | | B5 | 845–885 nm | 40 nm | 30 m | | |

*2.3. Mapping the ATOs, ACFOs, and MCFOs of Sentinel 2 and Landsat 8 Imagery over the APRFG*

Total and cloud-free observation maps are important for assessing the availability of Sentinel 2 and Landsat 8 data in APRFG. To create these maps, we analyzed the ATOs, ACFOs, and MCFOs of Sentinel 2 and Landsat 8. The process of calculating the ATOs, ACFOs, and MCFOs was implemented as shown in Figure 2. First, a customized binary band "B-cloudy" with a single value of '1' was created by duplicating any one of the existing bands to compute the total observations over the APRFG study area. This binary band was generated for all Sentinel 2 and Landsat 8 imagery collections. Additionally, another customized band, labeled 'B-cloudless' with single value of '1', was added to all satellite imagery collections to indicate the number of cloud-free observations. This band was created by using the QA60-based cloud-mask operation of Sentinel 2 and the CFMask of Landsat 8 data [38–40]. For Sentinel 2, the QA60 band contains information on whether the pixel is cloudy or not in 10th and 11th bit for opaque and cirrus clouds. For Landsat 8, the CFMask is a multi-pass algorithm that creates a cloud shadow mask by iteratively estimating cloud heights and projecting them onto the ground. Second, annual and monthly total and cloud-free observation maps were created by using a 'reduce (ee.Reducer.sum())' algorithm. Finally, the ATOs, ACFOs, and MCFOs were calculated and combined with an APRFG shapefile by using the '.reduceRegions' algorithm with

'ee.Reducer.mean()' reducer in the GEE environment. Through the above-mentioned steps, the shapefiles describing the total and cloud-free observations for each rice field in APRFG can be obtained.

The shapefiles of total and cloud-free observations from the Landsat 8 and Sentinel 2 images and statistical results were spatially displayed by using the ArcGIS platform (version 10.2). To further visualize the data availability during the key growth stages of rice over the APRFG, we furthermore visualized the MCFOs as box plot statistics. Considering the systematic overlapping of the scene paths of the satellite sensors, we also computed and mapped the areas of the APRFG whose ATOs were greater than the mean annual total observation (MATOs) according to the official revisit cycle (5 days of Sentinel 2: 73 MATOs; 16 days of Landsat 8: 23 MATOs).

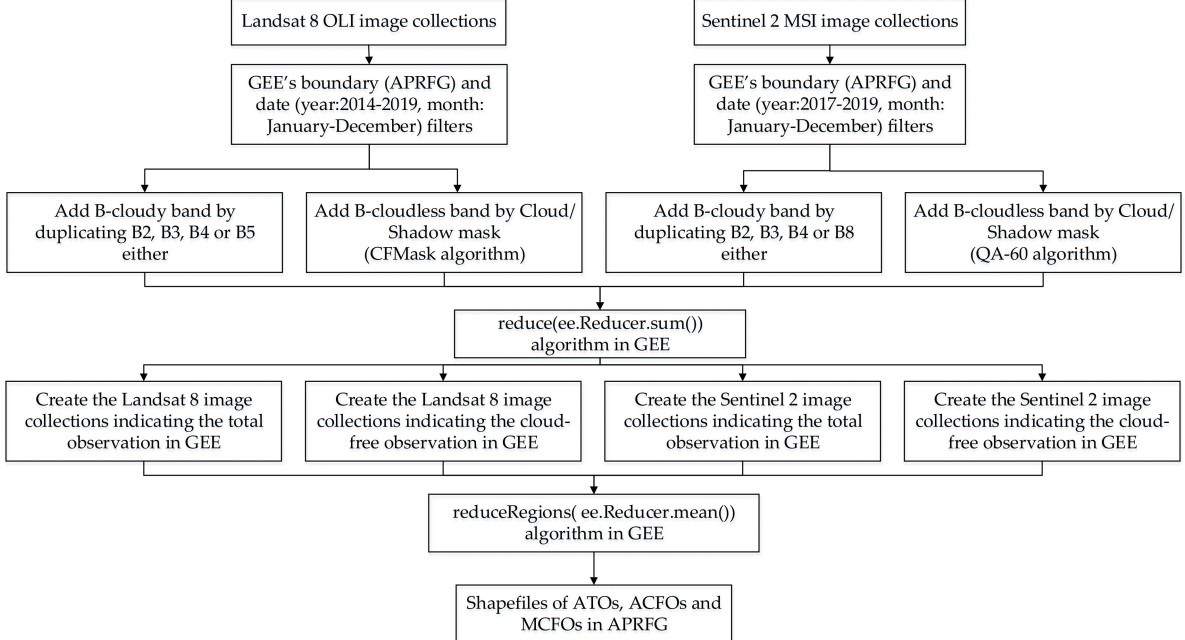

**Figure 2.** Records of the total and cloud-free observations of Sentinel 2 and Landsat 8 data over the APRFG.

### 2.4. Acquisition Probability (AP) Calculation of Cloud-Free Images

Acquisition probability (AP) of usable images is a useful indicator that not only stands for the probability of cloud cover in each paddy rice field, but also excludes the potential problem that the image overlap between satellite scenes/footprints may bring more total observations and cloud-free observations. The annual APs of ACFOs were computed by using Equation (1) over APRFG at the spatial scale of each paddy rice field. The computed APs from the Landsat 8 and Sentinel 2 images were then spatially displayed.

$$AP = \frac{CF}{Tot} \times 100\% \tag{1}$$

where *CF* represents the ACFOs for each paddy rice field, and *Tot* donates the ATOs for each paddy rice field.

### 3. Results

### 3.1. The ATOs Records of Sentinel 2 (2017–2019) and Landsat 8 (2014–2019) over the APRFG

The ATOs of the Sentinel 2 data in 2017–2019 and Landsat 8 data in 2014–2019 over the APRFG were shown in Figure 3. Figure 3a–f represents the ATOs of Landsat 8 from 2014 to 2019. The averaged ATOs of Landsat 8 from 2014 to 2019 were: 20.1 (2014), 21.2 (2015), 20.3 (2016), 21.3 (2017), 23.1 (2018), and 19.8 (2019). The median ATOs were 16, 18, 15, 17, 19, and 16, respectively, showing relatively stable observations. In addition, there are

many overlapping areas of adjacent along-orbit or cross-orbit images that were obtained at higher ATOs than other areas. The areas of the APRFG, shown in red in Figure 4a–f, are regions where Landsat 8 provided more than 23 MATOs. From 2014 to 2019, the grids of the APRFG (25,540 grids in total) whose ATOs were greater than 23 were 8224 (32.24%; 2014), 8171 (32.02%; 2015), 8285 (32.48%; 2016), 8239 (32.30%; 2017), 8521 (33.40%; 2018), and 7959 (31.20%; 2019), indicating that more than 30% of the rice fields in APRFG are located in the overlapping area of the satellite imagery.

Figure 3g–i displayed the unevenly distributed ATOs of Sentinel 2 in the APRFG from 2017 to 2019. The averaged ATOs of Sentinel 2 in the APRFG were: 49.3 (2017), 108.8 (2018), and 112.9 (2019), and the median ATOs are 33 (2017), 71 (2018), and 73 (2019). Similar to Landsat 8, the areas of the APRFG shown in red (see Figure 4g–h) have more than 73 MATOs mainly due to the overlap between the adjacent Sentinel 2 tiles. Detailed ATOs (mean, median, mode, max, min, and standard deviation (S.D.)) of Sentinel 2 and Landsat 8 are tabulated in Table 2. It can be seen that the ATOs mode of Landsat 8 and Sentinel 2 was 15–16 (2014–2019) and 70–71 (2018–2019), respectively, meaning that most paddy rice fields of the APRFG can obtain frequent 15–16 ATOs of Landsat 8 and 70–71 ATOs of Sentinel 2. The min and max ATOs also revealed that ATOs varied throughout the locations of paddy rice fields.

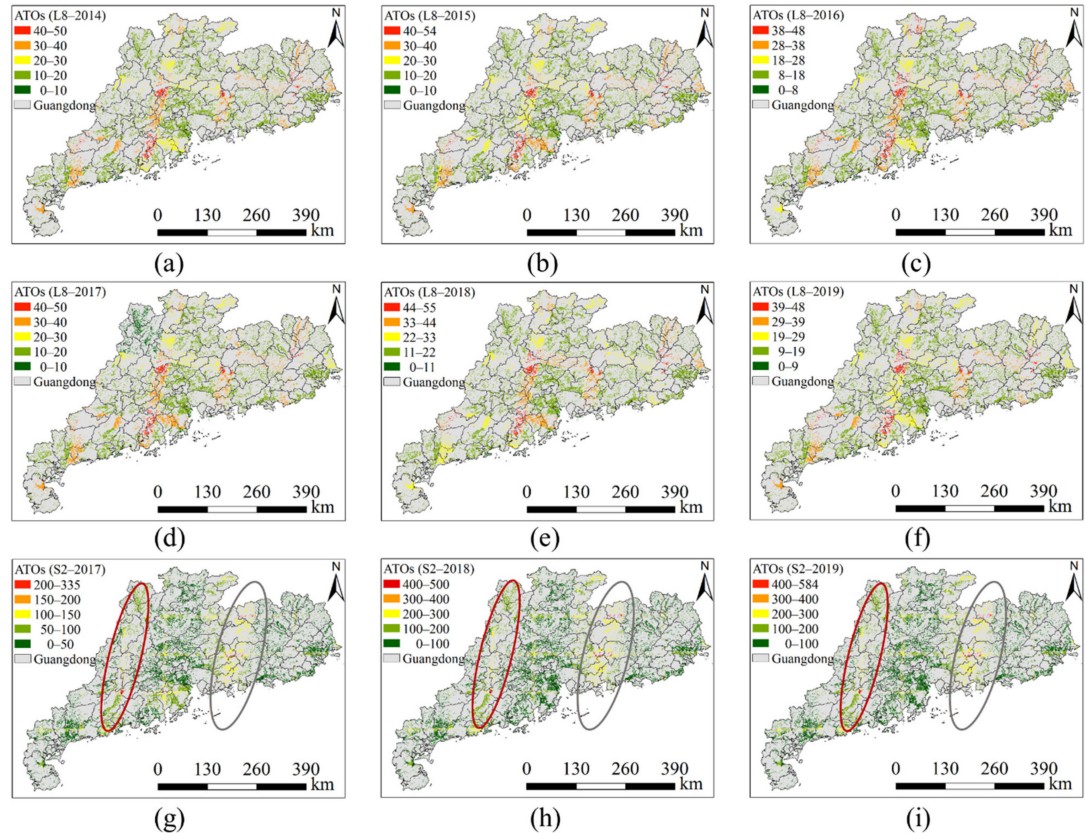

**Figure 3.** ATOs of Landsat 8 (L8 2014–2019; **a**–**f**) and Sentinel 2 (S2 2017–2019; **g**–**i**) over APRFG. The red and grey ovals implied that there are larger overlaps between UTM49 (tiles: 49RHJ−49RHE) and UTM50 (tiles: 50RKP−50RKK) of Sentinel 2 images.

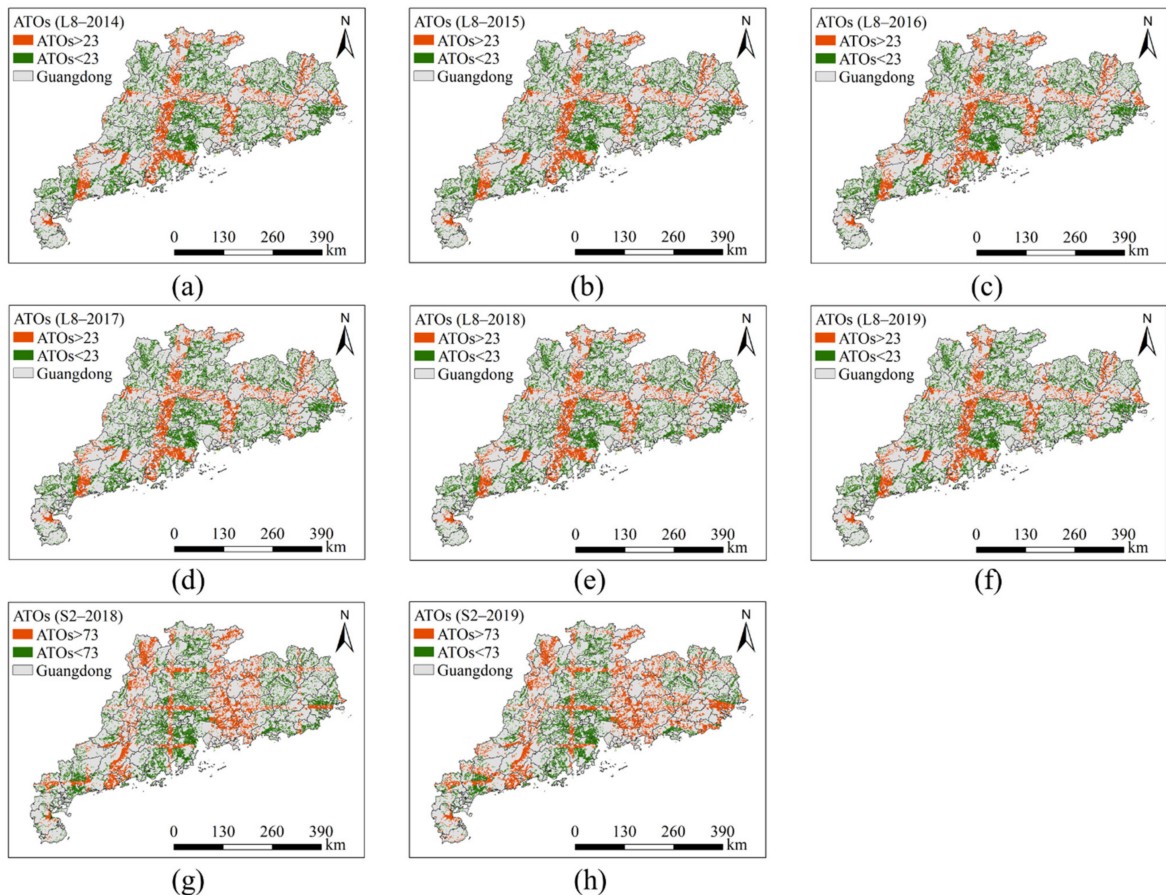

**Figure 4.** The regions whose ATOs were greater than the mean annual total observations (MATOs) in APRFG: (**a–f**) represented the ATOs (>23) of Landsat 8 from 2014 to 2019; (**g,h**) represented the ATOs (>73) of Sentinel 2 (2A + 2B) from 2018 to 2019.

**Table 2.** Annual total observations (ATOs) of the Landsat 8 and Sentinel 2 data over the APRFG.

| Platforms | ATOs | 2014 | 2015 | 2016 | 2017 | 2018 | 2019 |
|---|---|---|---|---|---|---|---|
| Landsat 8 | mean | 20.1 | 21.2 | 20.3 | 21.3 | 23.1 | 19.8 |
| | median | 16 | 18 | 15 | 17 | 19 | 16 |
| | mode | 15 | 16 | 15 | 15 | 15 | 15 |
| | max | 50 | 54 | 49 | 55 | 55 | 48 |
| | min | 11 | 9 | 11 | 10 | 12 | 11 |
| | S.D. | 8.8 | 9.3 | 8.8 | 9.4 | 10.3 | 8.5 |
| Sentinel 2 | mean | / | / | / | 49.3 | 108.8 | 112.9 |
| | median | / | / | / | 33 | 71 | 73 |
| | mode | / | / | / | 29 | 70 | 71 |
| | max | / | / | / | 335 | 563 | 584 |
| | min | / | / | / | 27 | 65 | 69 |
| | S.D. | / | / | / | 30.4 | 67.4 | 69.6 |

### 3.2. The ACFOs Records of Sentinel 2 (2017–2019) and Landsat 8 (2014–2019) over the APRFG

The ACFOs maps of Sentinel 2 and Landsat 8 are shown in Figure 5. In Figure 5g–i, the uneven distribution of ACFOs and the higher abundance in areas of orbital overlaps of Sentinel 2 scenes can be seen. Figure 5a–f showed the ACFOs of Landsat 8 over the APRFG from 2014 to 2019. Similar to Sentinel 2, the distribution maps showed varying cloud cover frequency over the study area, especially the areas with no overlap. To visualize the ACFOs in detail, we plotted the frequency histogram of the ACFOs of Landsat 8 and Sentinel 2 over the APRFG as shown in Figure 6. Figure 6a–f showed the frequency histogram of

the ACFOs of Landsat 8 from 2014 to 2019. From 2014 to 2019, the most frequent ACFOs ranges from 6 (13.49%: 3445 grids in APRFG; 2014), 5 (13.65%; 2015), 7 (16.72%; 2016), 6 (12.82%; 2017), 8 (12.36%; 2018), and 7 (13.57%; 2019). The averaged ACFOs of Landsat 8 were: 9.53 (2014), 8.50 (2015), 10.07 (2016), 9.46 (2017), 10.70 (2018), and 9.87 (2019).

Figure 6g–i represents the frequency histogram of the ACFOs of Sentinel 2 from 2017 to 2019 over the APRFG. From 2017 to 2019, the most frequent ACFOs ranges from 12–14 (4738: 18.56%; 2017), 25–30 (7746: 30.36%; 2018), and 30–35 (7190: 28.18%; 2019). The averaged ACFOs of Sentinel 2 in 2017–2019 were: 20.44 (2017), 43.34 (2018), and 48.28 (2019). Similar to Landsat 8, there are many areas of overlapping images and corner regions of adjacent images that received more ACFOs than other areas. In 2018 and 2019, such areas accounted for 39.94% and 48.16%, respectively, of the total area.

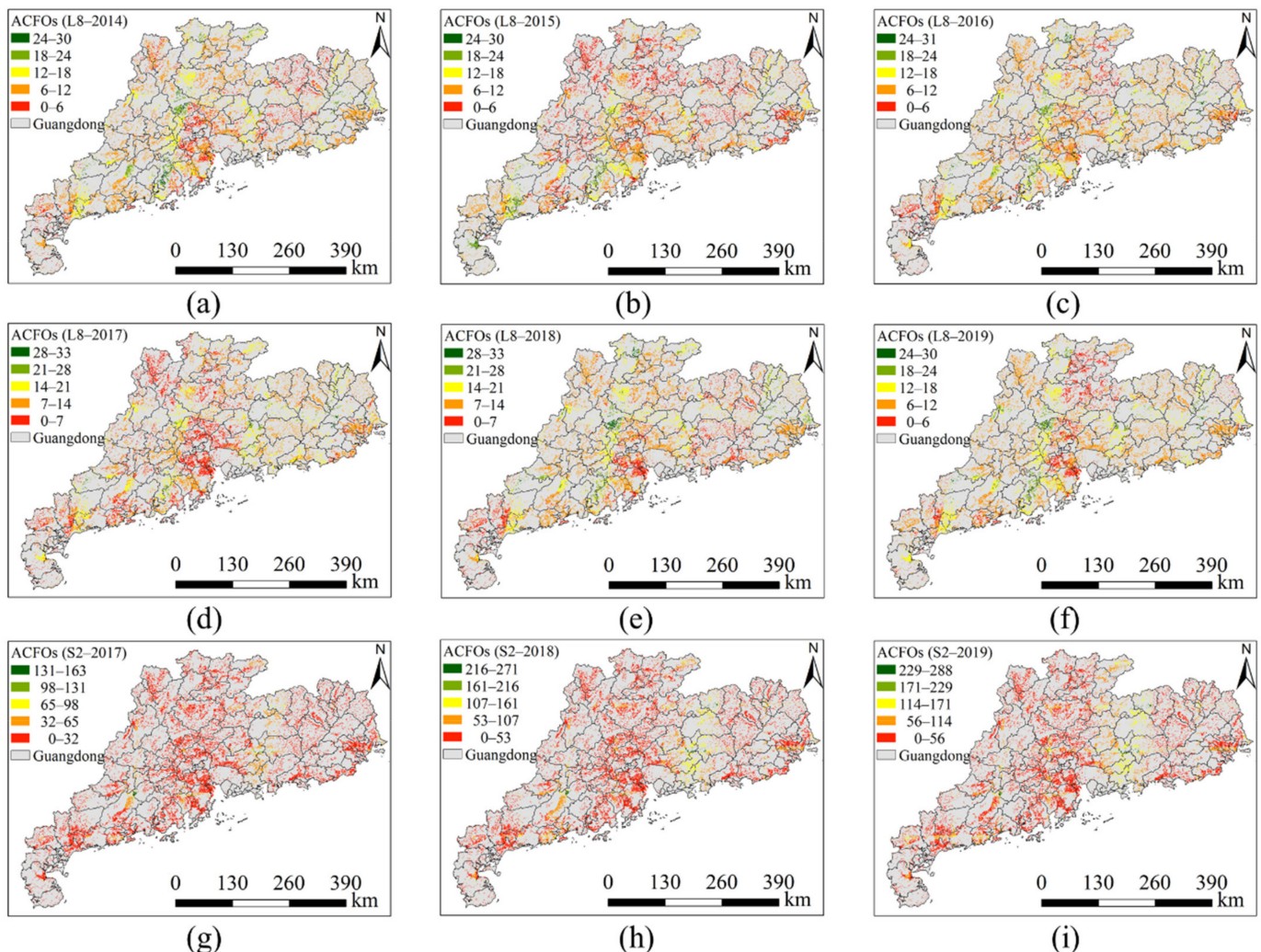

**Figure 5.** ACFOs of Landsat 8 (L8 2014–2019; **a**–**f**) and Sentinel 2 (S2 2017–2019; **g**–**i**) over APRFG.

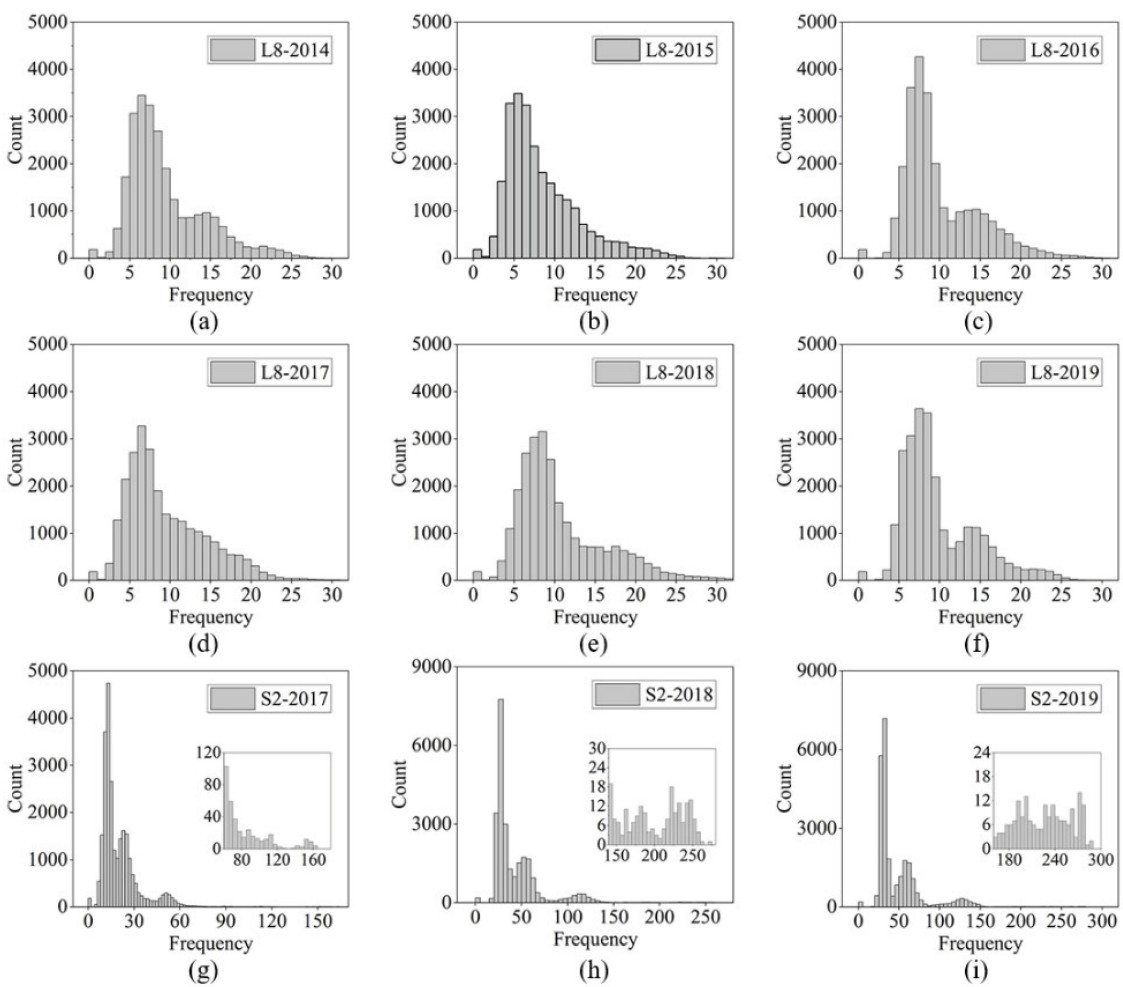

**Figure 6.** The frequency histogram of ACFOs of Landsat 8 in 2014–2019 (**a**–**f**) and Sentinel 2 in 2017–2019 (**g**–**i**) over the APRFG. Note: (**g**) showed the results of Sentinel 2 in 2017, in which the total count was less than in 2018–2019 as Sentinel 2B images were only first available in August 2017.

### 3.3. The MCFOs Records of Sentinel 2 (2017–2019) and Landsat 8 (2014–2019) over the APRFG

In rice production, demand for remote sensing products is seasonal. In practice, the growth stage of early rice in APRFG is from March to July, and the growth stage of late rice is from July to November, with five growing periods. Therefore, the availability of remote sensing products is more important from March to November than between December and February. To this end, we recorded the MCFOs of Sentinel 2 (2017–2019) and Landsat 8 (2014–2019) over the APRFG study area.

Figure 7a–c depicts the MCFOs of Sentinel 2 in APRFG from 2017 to 2019, showing the monthly fluctuating MCFOs. Due to lack of data from Sentinel 2B over the APRFG before August 2017, only records from 2018 and 2019 were performed for this analysis. The averaged MCFOs in 2018 were 5.03 (March), 1.81 (April), 4.80 (May), 3.24 (June), and 3.77 (July). The averaged MCFOs in 2018 were 2.86 (August), 3.77 (September), 5.25 (October), and 4.36 (November). The averaged MCFOs in 2019 were 2.63 (March), 1.74 (April), 1.83 (May), 2.57 (June), 2.69 (July), 4.87 (August), 6.59 (September), 6.27 (October), and 7.75 (November). Overall, the MCFOs were greater from August to November than from March to July, meaning that there are more data available to monitor late rice than early rice. Figure 7d showed the hot spot of MCFOs of Sentinel 2 over the APRFG. It can be seen that Sentinel 2 supports more frequent cloud-free observations over the study area due to its two-unit operating mode.

The amount of ACFOs of Landsat 8 over the APRFG was generally lesser than that of Sentinel 2 due to its comparatively lower revisit period. It can also be seen that the MCFOs varied monthly from 2014–2019 (Figure 8a–f) but, in general, the amount of MCFOs was greater between August and November than between March and July. Figure 8g shows the hot spot of the MCFOs of Landsat 8 over the APRFG. Due to the nominal 16-day revisit of Landsat 8, many paddy rice fields in the APRFG obtained relatively lesser MCFOs compared to Sentinel 2. The paddy rice fields whose early rice received no MCFOs accounted for 60.7% (2014), 50.77% (2015), 37.84% (2016), 56.97% (2017), 17.2% (2018), and 53.06% (2019) of the APRFG, whereas late rice which received no MCFOs were 14.09% (2014), 34.5% (2015), 16.12% (2016), 33.16% (2017), 34.61% (2018), and 13.12% (2019) of the APRFG. Overall, Landsat 8 provided more limited MCFOs in the APRFG compared to the Sentinel 2, especially for the growing stages of early rice.

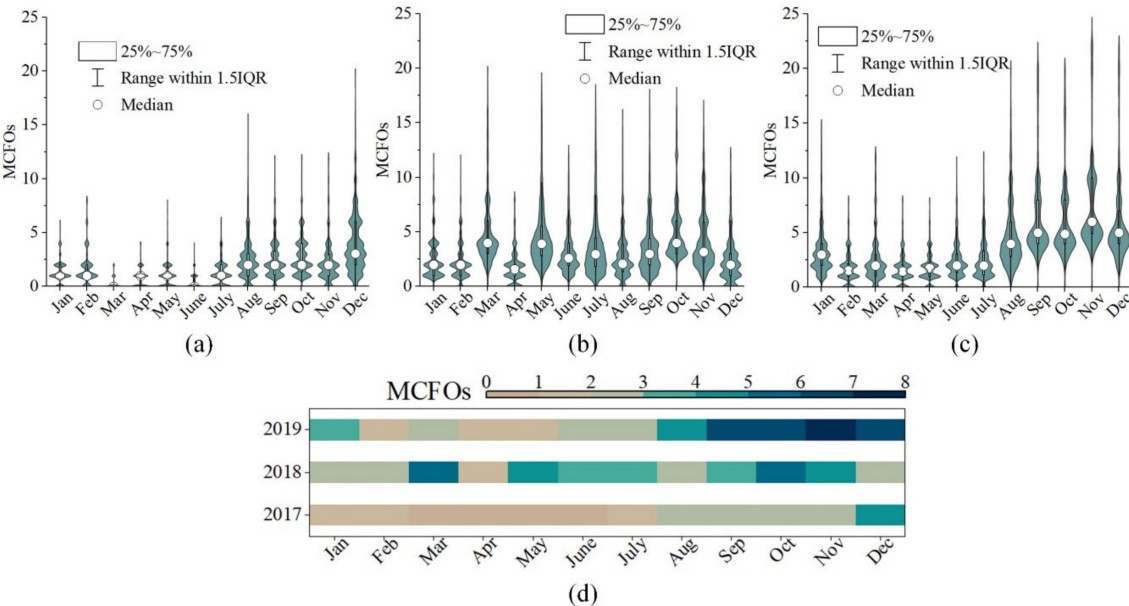

**Figure 7.** MCFOs of Sentinel 2 over the APRFG from 2017–2019: (**a–c**) the box plot and (**d**) the hot plot of the MCFOs. Note that there are no observations of Sentinel 2B before August 2017.

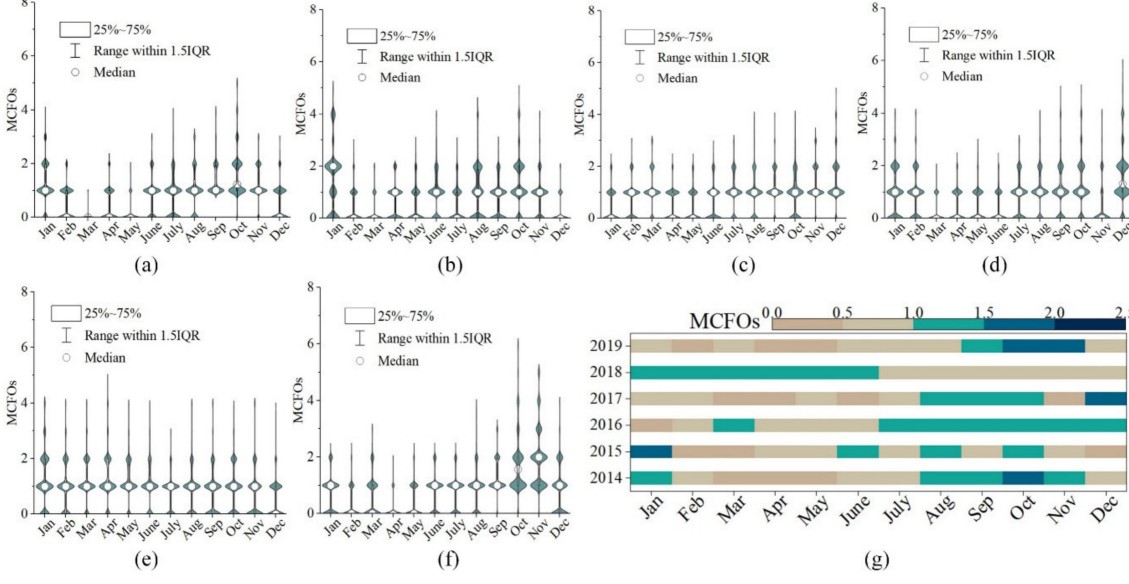

**Figure 8.** MCFOs of Landsat 8 over the APRFG from 2014 to 2019: (**a–f**) the box plot and (**g**) the hot plot of the MCFOs.

### 3.4. Spatial Differences of APs of ACFOs over the APRFG

The ACFOs and MACOs of Sentinel 2 and Landsat 8 showed cloudy frequency among all the paddy rice fields in Guangdong; however, the former statistics reflected only the cloud-free number of Sentinel 2- and Landsat 8-acquired imagery. In fact, redundant repeat observations of the numerous adjacent satellite images, which provided the same spectral information, are available for most areas of the APRFG in a single day. For rice monitoring, the time-series remotely sensed data is useful for the VIs inversion and therefore, the APs of ACFOs is paramount and indeed meaningful for observing the rice growth. The APs of Landsat 8 and Sentinel 2 shown in Figure 9 vary with position, time, and sensors. Moreover, it can be seen in Figure 10 that the median line and the mean of APs both of Landsat 8 and Sentinel 2 were less than 50%, in spite of some outliers. Table 3 describes the annual mean and median of APs of Landsat 8 and Sentinel 2. In detail, the mean APs of Landsat 8 were less than 52.1% and that of Sentinel 2 were less than 47.7%. Although the ACFOs and MCFOs reported above for Landsat 8 were smaller than those for Sentinel 2 due to the inherent revisit cycle, the APs for Landsat 8 were larger overall than those for Sentinel 2. Therefore, the APs provided an accurate description of the limited availability of ACFOs of Landsat 8 and Sentinel 2 over the AFRFG.

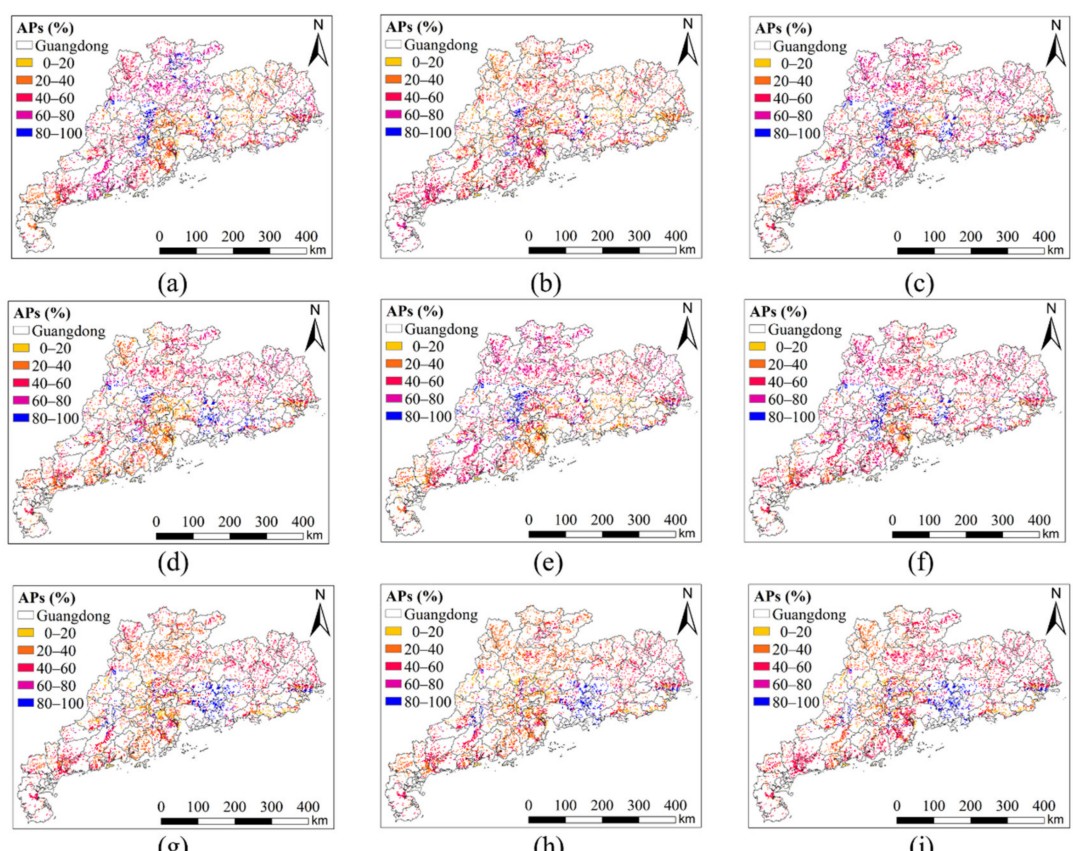

**Figure 9.** The APs distribution of ACFOs over APRFG: (**a**–**f**) the APs distribution of Landsat 8 from 2014 to 2019, (**g**–**i**) the APs distribution of Sentinel 2 from 2017 to 2019.

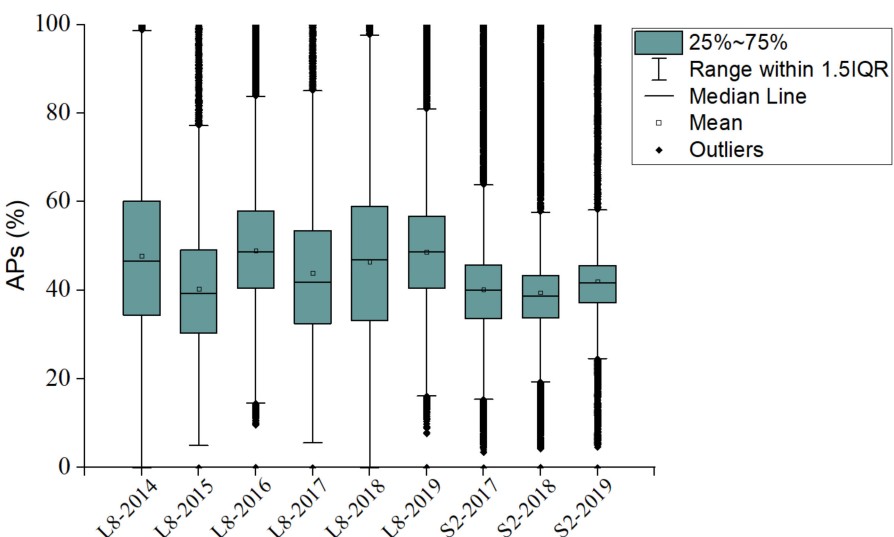

**Figure 10.** Box plot of APs of Landsat 8 and Sentinel 2 over the APRFG.

**Table 3.** Mean and median APs of ACFOs of Landsat 8 and Sentinel 2 over the APRFG.

| APs | L8−2014 | L8−2015 | L8−2016 | L8−2017 | L8−2018 | L8−2019 | S2−2017 | S2−2018 | S2−2019 |
|---|---|---|---|---|---|---|---|---|---|
| Mean | 49.9 | 41.1 | 51.7 | 46.4 | 49.9 | 52.1 | 46.2 | 44.4 | 47.7 |
| Median | 47.2 | 39.4 | 49.3 | 42.6 | 47.7 | 49.3 | 40.5 | 39.0 | 42.0 |

## 4. Discussion

Rice monitoring based on optical satellite sensors (passive sensors) in tropical and subtropical areas is always a challenge. Reduced Sentinel 2 and Landsat 8 data availability, caused by cloud cover or other impediments, may result in the failure of the time-series monitoring for rice [41]. This problem is obvious in regions with double or multiple rotation agricultural systems, where cropland tends to be covered by plants year-round, thus requiring high-frequency remote sensing monitoring [42]. This study considered the rainy and cloudy climate characteristics of the Guangdong province that can significantly influence the usability of Sentinel 2 and Landsat 8 data. Moreover, ATOs, ACFOs, and MCFOs, and maps and APs of ACFOs of Sentinel 2 (2017–2019) and Landsat 8 (2014–2019) over the APRFG were computed to quantitatively assess the availability of imagery from these two satellite platforms and to answer the issue of 'observability'.

### 4.1. The Total and Cloud-Free Observations of APRFG of Sentinel 2 and Landsat 8

For the purposes of rice growth monitoring, it is reasonable to assume that most satellite sensors have a limited observability of cloud-free imagery. Importantly, the frequent cloud cover over subtropical study areas such as the APRFG can aggravatingly impede the efficient use of optical satellite imagery. According to our results, described in Sections 3.1 and 3.2, some areas in APRFG had a greater cloud-free satellite image coverage than other areas during the study period. The areas with highest ATO and ACFO frequency were mainly located in the overlapping and corner regions of adjacent images (Figure 4). This can be explained by the swath overlap between adjacent orbits which result in some areas being observed from more than one orbit per revisit cycle. Additionally, such overlaps increase with latitude [43]. In detail, due to this 10 km overlap [44] of the Military Grid Reference System (MGRS) tiles projected in the Universal Transverse Mercator (UTM), these areas have more observations than other areas (Figure 4g–h). On the other hand, the APRFG spans the 114° E longitude which is located in the demarcation line of UTM zones 49 and 50. This implies that there are larger overlaps between UTM49 (tiles: 49RHJ−49RHE) and UTM50 (tiles: 50RKP−50RKK) and, thus, these areas had a higher coverage. It can be seen that there are areas, marked in red and grey ovals in Figure 3g–i,

that had more ATOs. Different with Sentinel 2, Landsat 8 acquires about 740 scenes a day on the Worldwide Reference System−2 (WRS−2) path/row system, with a swath overlap (or sidelap) varying from 7%at the equator to a maximum of approximately 85% at extreme latitudes. As seen in Figures 3a–f and 4a–f, similar to the grid pattern of Sentinel 2, there are some fields located in the overlapping area of the satellite images that have more ATOs and ACFOs than other fields. Additionally, the variation of ACFOs of Landsat 8 and Sentinel 2 imagery may also be influenced by the location of paddy rice fields, date, satellite sensors, revisit circle, regional and global climate, and water vapor cycle, etc. Most of the natural factors mentioned above are not controllable, and the visible factors affecting ACFOs are the satellite sensors and the corresponding revisit cycles.

Furthermore, the frequency histogram of the ACFOs of satellite imagery in APRFG (Figure 6) suggested the study area mostly has limited ACFOs. More specifically, the MCFOs of Sentinel 2 (Figure 7) and Landsat 8 (Figure 8) suggested that the early rice period had a higher probability of MCFOs than the late rice period. This is interpreted to be mainly due to the characteristic climate patterns of Guangdong, where highest amounts of precipitation occur on the first half of the year [45], coinciding with the late rice period. In particular, the high cloud coverage observed for the summer period may jeopardize crop-monitoring systems based on optical remote sensing imagery in tropical areas [46].

In fact, the cloud cover of Landsat 8 scenes can be minimized by using Landsat-MODIS image fusion to produce daily Landsat-resolution synthetic data for crop monitoring using STARRM and ESTARFM fusion models [47,48]. For Sentinel 2, the Harmonized Landsat Sentinel−2 (HLS) has been established and widely used in many cases. However, STARRM and ESTARFM models need either Landsat 8 or MODIS to have cloud-free observations at the same time. For example, in areas such as Guangdong, the presence of consistent cloudy days may affect the availability and accuracy of the models. When remote sensing data are needed for critical growth periods of rice, the cloud affects the use of both independent and fused data. While the HLS product combined the Landsat 8 and Sentinel 2 data, this dataset is not globally available (excluding places such as Guangdong) at this time, and is continuously being updated [49]. On the other hand, it is reported that it is hard to assess land-cover change in a region when cloud cover is consistently 30% or more [50]. In our study, the mean APs of Sentinel 2 and Landsat 8 were 52.1% and 47.7%, respectively, meaning that it is possible to conduct the time-series observation of rice. However, it can be concluded that it is difficult to provide in-time cloud-free satellite-acquired imagery when the remotely sensed data are urgently required. Compared to the study by Henri et al. [26] in the cloud-related analysis, which is the limited temporal scope (a single year) that not permitted to take into account the inter-annual variability of ACFOs, we simultaneously used several years of data to create the ACFOs and MCFOs map for Sentinel 2 and Landsat 8.

### 4.2. Methods for Improving the Usability the Sentinel 2 and Landsat 8 Images

Overall, we still suggest using free satellite data to monitor rice growth. Nevertheless, in places where the weather conditions limit the number of cloud-free days, satellite data may be insufficient for monitoring rice growth. In this context, the ability to supplement satellite data by means of other datasets is also of paramount importance. One way to fill the gaps left by cloud coverage is to complement satellite data using other sensors [51]. Therefore, conversion functions need to be developed to convert spectral data from Landsat 8, Sentinel 2, and other sensors to mitigate inconsistencies in surface reflectance and vegetation indices due to spatial resolution and spectra configuration [52]. Nowadays, unmanned aerial system (UAS) platforms' low dependence on cloudless conditions, high mobility during data acquisition, controlled revisit cycles, and extremely high ground resolution make them an excellent complement to satellite imagery [53,54]. In particular, most of the world's rice paddies are located in areas with continuous cloudy conditions, and therefore satellite data can be less available than for other crops [55,56]. On the other hand, in the field of crop monitoring, optical remote sensing is more sensitive to environmental

factors than, for instance, microwave remote sensing. The use of optical remote sensing technology alone may not always meet the needs of crop monitoring in crop production areas where cloud cover is often present. With radar-signal insensitive to cloud cover and illumination conditions, synthetic aperture radar (SAR)-based techniques are widely used as practical options for mapping the area and spatial distribution of paddy rice. As VIs are now a proven, well-observed proxy for estimating crop growth, there is a need to model a combination of VIs and radar-derived information [57–59]. In addition, the integration of active sensors into the monitoring rice growth could be an option. Several studies showed the potential of different non-imaging active sensors for crop monitoring, while the imaging active sensors that support more pixel-level spectral information remains to be solved [60–62].

### 4.3. Limitations and Prospects

In this study, the maps of ATOs, ACFOs, and MCFOs created by using a 1 km × 1 km grid cell size potentially led to a few inaccuracies. In the future, we recommend conducting similar studies with higher spatial resolution to avoid any such losses. For example, the global 30 m land use classification data (1980–2018) is available from 2020 onwards ( http://data.casearth.cn/, accessed on 1 June 2021). In spite of relative course resolution, the cloud-related results in the study illustrated the impact of cloud cover on Sentinel 2 and Landsat 8 optical observations in the study area. Another shortcoming of the experiment is that only two data sources were compared, mainly as they are freely available compared to other sources that have finer ground sampling distance. In addition, useable remote sensing satellites with higher resolution (e.g., the China–Brazil Earth Resource Satellites, HJ−1A and HJ−1B environment and disaster reduction satellites, and high-definition Earth observation satellite GF−8), could be used for analyzing the cloud cover [63].

### 5. Conclusions

In subtropical areas such as Guangdong, China, the large amount of remote sensing data provided by the optical satellite sensors (e.g., Landsat 8 OLI and Sentinel 2 MSI) could have provided new opportunities to observe VIs for rice at a large scale, while the cloud cover reduced the chances of cloud-free observation for rice. In order to quantitatively analyze the influence, we presented ATOs, ACFOs, MCFOs, and ACFO-related APs to elaborate on the probability of satellite data (Landsat 8 and Sentinel 2) for acquired the un-contaminated imagery for managing rice in Guangdong Province, China. The quantitative evaluation was summarized as follows:

1) The ATOs of Landsat 8 showed relatively stable observations compared to the Sentinel 2, and the per-field ACFOs of Sentinel 2 and Landsat 8 were unevenly distributed;
2) The MCFOs of Sentinel 2 and Landsat 8 were greater between August and December than between January and July. As the second half of the year coincides with the late rice season, there is more data available to monitor late rice than early rice;
3) The AP of usable ACFOs with 52.1% (Landsat 8) and 47.7% (Sentinel 2) indicated that these two satellite sensors provided markedly restricted observation capability for rice in the study area.

Our results suggested that the availability of the cloud-free observations varies as a function of the location, date, and sensor platform, showing the limited applicability of optical remote sensing for observing rice in Guangdong.

**Author Contributions:** R.J. was responsible for the framework design of the entire project in this research and arranged tests, conducted several verification tests, and wrote the article. J.H. and Y.X. assisted in collected all the test data and checked the paper carefully. Z.Z. and Y.Z. proposed the main plans, ideas, and guidance for the work and reviewed the paper, as well as acquired the funding. X.L., X.C., A.S.-A. and K.L. provided guidance and advice. All authors have read and agreed to the published version of the manuscript.

**Funding:** This research was funded by [National Natural Science Foundation of China] grant number [31871520], [China Scholarship Council] grant number [201908440388], [Guangdong Basic and Applied Basic Research Foundation] grant number [2020A1515110214], [National Key R&D Program of China] grant number [2018YFD0200301], [Project of Rural Revitalization Strategy in Guangdong Province] grant number [2020KJ261], [Science and Technology Plan of Guangdong Province of China] grant number [2021B1212040009], [Innovative Research Team of Agricultural and Rural Big Data in Guangdong Province of China] grant number [2019KJ138] and [Open operation project of Key Laboratory of new rice breeding technology in Guangdong Province] grant number [2017B030314173]. And the APC was funded by [National Natural Science Foundation of China] grant number [31871520].

**Data Availability Statement:** The data presented in this study are available in the article.

**Acknowledgments:** The authors would like to thank the reviewers and editors for their constructive comments. We also thank AJE (https://secure.aje.com/) for its linguistic assistance during the preparation of this manuscript.

**Conflicts of Interest:** The authors declare no conflict of interest.

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
