# Peer review of "Cloud Cover throughout All the Paddy Rice Fields in Guangdong, China: Impacts on Sentinel 2 MSI and Landsat 8 OLI Optical Observations"

_remotesensing, doi:10.3390/rs13152961_

Round 1
Reviewer 1 Report
Cloud cover is very important to all the paddy rice fields. This paper shows the quantified impact of clouds on satellite observations by mapping ATOs, ACFOs, and MCFOs. This is very interesting research in the optical satellite-acquired imagery.
I have some questions about this paper as follows
1. In section 2, Could you add the flowchart about the data process? I don’t clearly understand your work.
2. In the Figure 3 and 4, you need the detailed information about the reasons of difference of ATOs for each satellite.
3. In section 3.2, you simply described the variation of ACFOs in Figure 5 and 6. Could you explain more reasons for the variation?
4. In section 4, the discussion was totally redundant. So you rewrite the core issue in the discussion section.
Reviewer 2 Report
Very interesting work.
I have only a couple of comments:
The abstract should be shortened.
How is defined the threshold between cloud and cloudless? Often we have condition of low thin clouds which are more critical for some wavlengths and less critical for other.
A grid pattern is apparently visible (due to overlaps, see attached file): the authors should make some discussion on it .

Reviewer 3 Report
MS: Cloud cover throughout all the paddy rice fields in Guangdong, China: Impacts on Sentinel 2 MSI and Landsat 8 OLI optical observations
The manuscript utilizes Sentinel 2 MSI and Landsat 8 OLI imagery to compare rice field areas in Guangdong, China under two cloud cover scenarios. The topic is widely interesting to tropics and covers certain aspect of remote sensing. However, there is a major drawback of the methods which lacks information to follow the steps. The conclusion is not much supported from the study.
Please address the following minor comments:
Title: I feel paddy and rice are the synonyms. I looked up published articles to see whether ‘paddy rice fields’ are common usage. I found several articles refer paddy rice fields, so I will leave it up to you using paddy rice fields, paddy fields or rice fields either.
L 48: Rice fields or paddy field either
77: space between &Sentinel
L 131- L138: The authors indirectly mention objectives here, and use proper words like goals, objectives, aims to indicate objectives.
L 139: Methods
L 143: Space between number and Unit - 22.3°C
L 162 -185: I did not see the information on how many images used for both Sentinel 2 and Landsat 8 for the analysis
L 169: The authors used several terms i.e. ATOs, ACFOs, and MCFOs without giving proper definitions and detail information. There are certain thresholds used when presenting results i.e. ATO, but I did not see authors justify why those specific cutoff points. So please give definition/addition information.
Authors adopt mask and unmasked data for the analysis and refer the methodology using some references (L 181-184). Giving some addition information for those methodologies is useful to readers to fully understand the methods.
L 253: Authors tabulate basic univariate statistics including mean, min, max etc. I was wondering how the results of standard deviation correlate with the above statistics.
L 260: Give descriptive figure caption. What are the circles indicate from the Sentinel 2 images
L 325-326: Give descriptive caption about a-d.
Conclusion: Too general and not remarkable findings stated.
Round 2
Reviewer 3 Report
Good revision and the revised manuscript addressed the most comments suggested.